

# Have trends changed over time? A study of UK peak flow data and sensitivity to observation period.

Adam Griffin, Gianni Vesuviano, Elizabeth Stewart

Centre for Ecology & Hydrology, Wallingford, Oxfordshire, OX10 8BB. UK.

*Correspondence to*: Adam Griffin (adagri@ceh.ac.uk)

**Abstract.** Classical statistical methods for flood frequency estimation assume stationarity in the gauged data. However, recent focus on climate change and, within UK hydrology, severe floods in 2009 and 2015 have raised the profile of statistical analyses that include trends.

This paper considers how parameter estimates for the Generalised Logistic distribution (standard for UK annual maximum

flows) vary through time, using the UK Benchmark Network (UKBN2) to separate the effects of land-use change from climate change. We focus on the sensitivity of parameter estimates to adding data, through fixed-width moving window and fixed-start extending window approaches, and on whether parameter trends are more prominent in specific geographical regions.

Under stationary assumptions, the addition of new data tends to further the convergence of parameters to some "final" value. However, addition of a single new data point can vastly change non-stationary parameter estimates. Little spatial correlation

is seen in the magnitude of trends in peak flow data, potentially due to the spatial clustering of catchments in the UKBN2. In many places, the ratio between the 50-year and 100-year flood is decreasing, whereas the ratio between the 2-year and 30-year flood is increasing, presenting as a "flattening" of the flood frequency curve.

## 1 Introduction

Over the last decade, the United Kingdom has seen several extreme flood events, particularly as a result of significant winter

storm events in 2009 and 2015-16 (Barker, *et al.*, 2016; Defra, 2016). The 2015-16 storms took place over the Lake District in north-west England, and during the event record observations of 24-hour and 2-day rainfalls were seen (Marsh, *et al.*, 2016; Spencer *et al.*, 2018). This has added weight to various questions about whether this frequency of extreme events is indicative of some change in the nature of the flooding due to changes in rainfall patterns as a result of climate change, or due to land use changes and river channel alterations (IPCC, 2014). Within statistical flood frequency estimation, one common assumption is

that the time series of annual maxima or threshold exceedances (peaks-over-threshold) is stationary: the underlying modelling distribution is constant in time. However, this may not be wholly appropriate in all cases. Taking this non-stationarity into account may be crucial in flood risk management (Reynard, *et al.*, 2017) due to the potential for underestimates of reliability of defence structures, or over-spending due to the failure to account for a reduction in flood estimates. Spencer *et al.* (2018) also use up-to-date NRFA data to look into whether the record-breaking events are reason for practitioners to adopt non-





stationary assumptions, highlighting historical data and local data as ways to supplement the existing data, being used as evidence for trends and to improve associated uncertainties.

Many authors have tried different approaches to the study of trends and non-stationarity in river flow data and have investigated how to apply statistical modelling to the problem. Typically, it is difficult to disentangle the effects of land-use change and

climate in river flow regimes, due to simultaneous changes in both. Hannaford and Marsh (2006) developed a hydrological reference network, the UK Benchmark Network (explained below), to analyse changes in river flow in locations without anthropogenic influence. Harrigan *et al.* (2017) used the updated UK Benchmark Network to study high flow and low flow trends, looking at 5[th] and 95[th] percentiles of daily discharge data. Hall *et al.* (2014) have investigated flood regime changes on a European scale to identify possible generating mechanisms, and the current methods of observing or modelling these changes.

It can be challenging to make conclusions on long-term trends or the magnitude of long return period floods in the presence of short record lengths in many locations, so various statistical approaches have been brought forward. O'Brien and Burn (2014) use several extreme value distribution parameters to estimate trends in peak flow in Canada, using parameters which evolve linearly in time; regionalisation was also implemented using trend direction as a pooling criterion. Prosdocimi *et al.* (2014) use a 2-parameter Log-normal distribution to analyse trends in UK peak flow data using time and annual 99[th] percentile

of daily rainfall as covariates, to account for the fact that trends may not be linear in chronological time, but may be relative to changes in precipitation. Kay and Jones (2012) apply isotonic regression to look for monotonic changes in flood frequency in Britain. More recently Eastoe (2019) used a random effects model across the UK using peak-over threshold data. Future Flows Hydrology is a UK-nationwide probabilistic hydrological projection of trend using deterministic hydrological models to compare projections to baseline (1961-1990) high flow and low flow behaviour and to analyse the associated uncertainty

(Collet *et al.*, 2018).

One problem in the estimation of flood frequency in the presence of non-stationarity is that single significant events can have a great effect on estimates of flood magnitude and uncertainty estimates, which is compounded under trends. For example, actually observing the "1-in-1000-year" flood in a 40-year monitoring period may lead to overestimation in the upper tails of the flood frequency curve. In related work, Kjeldsen and Prosdocimi (2016) found no clear drivers behind the most "surprising"

events, those much bigger than any in the current record, which overwhelmed defences in the UK.

Here, moving window and extending window methods are used with non-stationary formulations of the Generalised Logistic distribution (GLO) to highlight sensitivity in parameter fitting to record length. The aims of the paper are to:

- Investigate how flood frequency estimations change over time as records are extended
- Investigate how sensitive the parameters of the GLO are to the most extreme events.

- Demonstrate examples of issues in consistently describing changes in flood frequency estimates in the UK.

Section 2 will describe the NRFA data being used, the UK Benchmark Network and the Generalised Logistic distribution. Section 3 will outline the results of moving window and extending window analyses. In Section 4, results will be discussed, an explanation of the findings offered, and possible applications and extensions for this work suggested.





## 2 Data and methodology

### 2.1 Data

This study will focus on data from the National River Flow Archive (NRFA, 2018), and in particular on the UK Benchmark Network (UKBN2) (Harrigan *et al.*, 2017). An initial version was set up by Hannaford and Marsh (2006) to provide a collection

of near-natural catchments within the UK which have natural flow regimes broadly representative of the region, with high-quality hydrometric data. This dataset was updated in 2017 and has been used in the past to analyse trends in high and low flows in the UK. The current version has stations with between 21 and 86 years of record, with a mean length of 46 years. On average, these stations have 1.5% missing daily data.

For the present work, a subset of the data (73 stations) is used, consisting of UKBN2 stations that have 30 or more years of

annual maxima and are considered by the NRFA as "suitable for pooling"; see Fig. 1 for locations. This means that the three largest recorded AMAX values at a given station are likely to be close to their true value (NRFA, 2018). Due to the requirement of UKBN2 that catchments must be free of significant land use change over the period of record, catchments in the south-east and midlands of England are fewer in number and typically smaller than catchments located elsewhere. For some portions of the current work, the 73 catchments are further divided into those with 40 or more years of annual maxima (67 catchments)

and 50 or more years of annual maxima (29 catchments).

### 2.2 Methods

This paper focuses on how flood frequency estimates change over time as records are extended. To this end, the Generalised Logistic Distribution (GLO) is fitted using L-moments (Hosking and Wallis, 1997) to the AMAX series of peak river flow based on 15-minute readings for stations in UKBN2. This is done using both stationary parameters and non-stationary

parameters – values that vary over time – separately. These fitted parameters, along with estimates for the 1-in-30, 1-in-50 and 1-in-100 year floods, are compared spatially and temporally across the UK, applying moving fixed-width windows and extending fixed-start windows to the AMAX series.

In the UK, the Flood Estimation Handbook (FEH; Robson and Reed, 1999) states that the recommended distribution for the AMAX series is the Generalised Logistic distribution (GLO), given by the quantile function describing flow $Q$ (measured in

$m^3$/s) for return period $T$ (measured in years):

$$Q_T = \begin{cases} \xi + \dfrac{\alpha}{\kappa}(1 - (T-1)^\kappa) & \text{if } \kappa \neq 0 \\ \xi - \alpha \log(T-1) & \text{if } \kappa = 0 \end{cases}$$

with location parameter $\xi$, scale parameter $\alpha$, and shape parameter $\kappa$ (Hosking and Wallis, 1997). Note that QMED and $\xi$ are similar but subtly different: QMED is the median of the observed series, whereas $\xi$ is the median of an infinite series drawn from the same GLO distribution; Fig. 2 shows some examples of GLO flood frequency curves for different values of $\alpha$ and $\kappa$.

Under stationary conditions $T$ is equivalent to the annual exceedance probability (AEP) where $AEP = 1/T$.


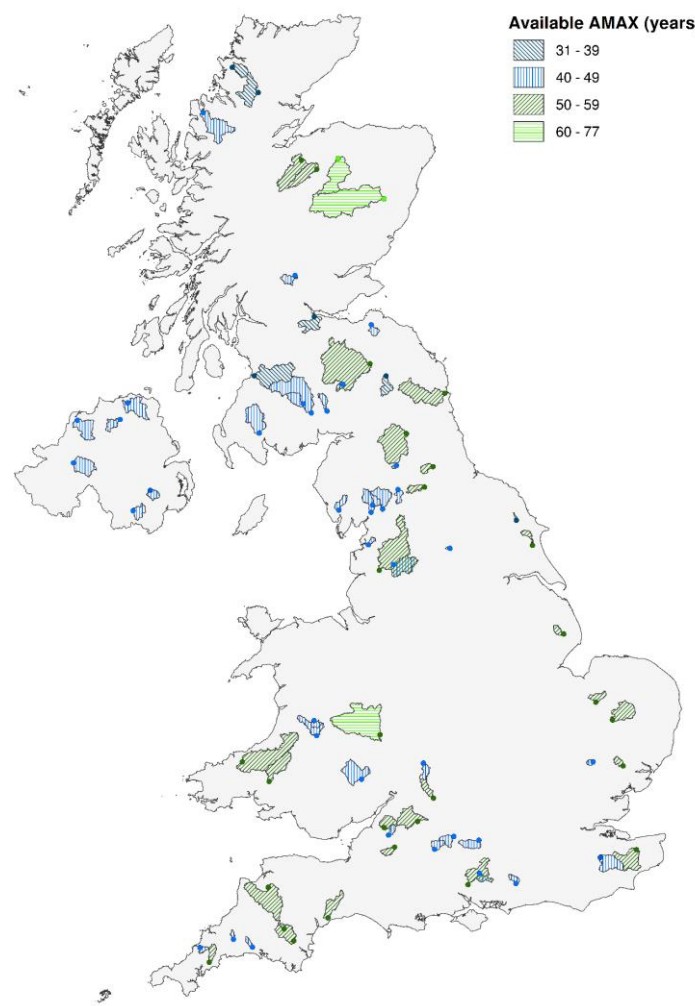

**Figure 1: Locations of the 73 stations used in the analysis, highlighting record length and location of associated watershed.**

## 3 Results

### 3.1 Moving window analysis

5   To begin with, this study uses a "moving window" analysis, which can be thought of as a "window of recent memory". Although it may not be reasonable to assume stationarity over the whole length of a given station's record, it may be reasonable to choose a small window during which there is no statistically significant trend. In particular, the identification of flood rich or flood-poor periods, investigated in Europe by Hall *et al.* (2014), may be a strong application for this method.

A fixed-width window of 20 years is applied to each record that has more years of AMAX data than the width of the window.
10  The window is moved across the record, year-by-year, from the start to the end. At each position, stationary GLO parameters




are fitted and the value of QMED is computed for only the AMAX data inside the window. This is repeated using a 30-year and 40-year window.

Comparisons between the time-series of parameters for the three different window sizes show that wider windows result in more lag (delay in time between the extreme event and an equivalent change in parameters) and attenuation ("smoothing out")

of changes to the parameter estimates, as the window is moved. The increased attenuation observed for wider windows is to be expected, as the largest event in a 20-year window has greater weight in parameter calculation than if it were the largest event in a 30 or 40-year window. The increased lag observed for wider windows can be explained as events from further back in the time-series taking longer to drop out of the window. However, the width of the window ultimately has little effect on the general overall trends observed. For this reason, only the 20-year window will be used in the rest of this paper to best

highlight differences between the start and end of records.

From a hydrological perspective, a distribution based on an AMAX record in which just one event is much larger than QMED (and many smaller) will have strongly negative $\kappa$, while a record with several similarly sized events much larger than QMED (and few smaller) will result in a strongly positive $\kappa$, which could also suggest a possible maximum flow rate at the station. Hence, for moving window analyses, the change in the shape parameter over time relates to the introduction and, in some

cases, later ejection of events either much larger or much smaller than QMED. This can be seen at the end of the example in Fig. 3, where the extreme event (the largest in the record) creates a great change in the moving window estimate of the shape parameter.

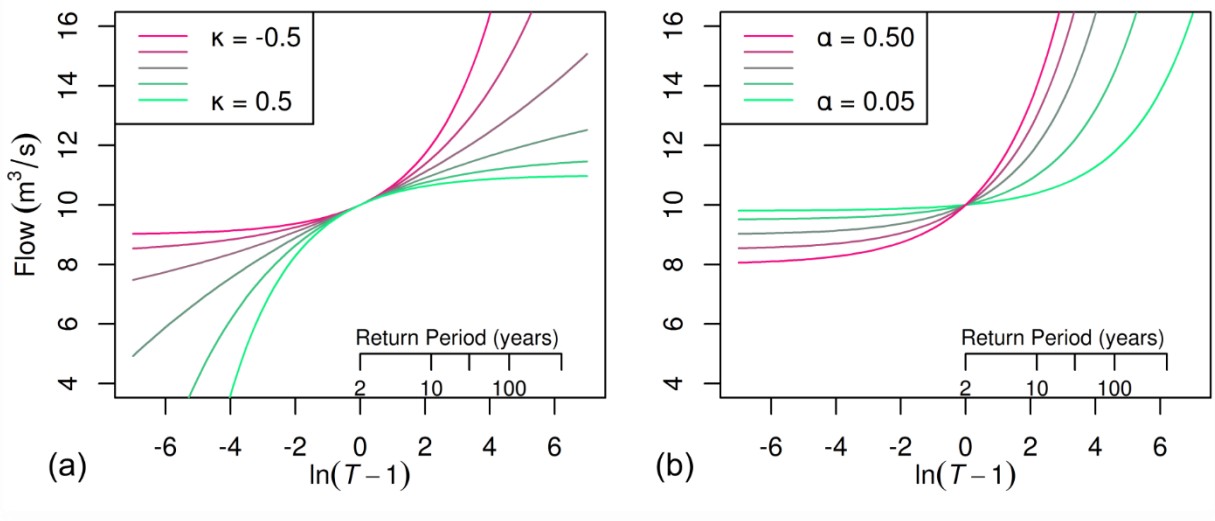

**Figure 2: Example GLO flood frequency curves. (a): varying $\kappa$, with $\alpha$=0.5, $\xi$=10; (b): varying $\alpha$ with $\kappa$=0.5, $\xi$=10. Plotted on logistic reduced variate scale.**



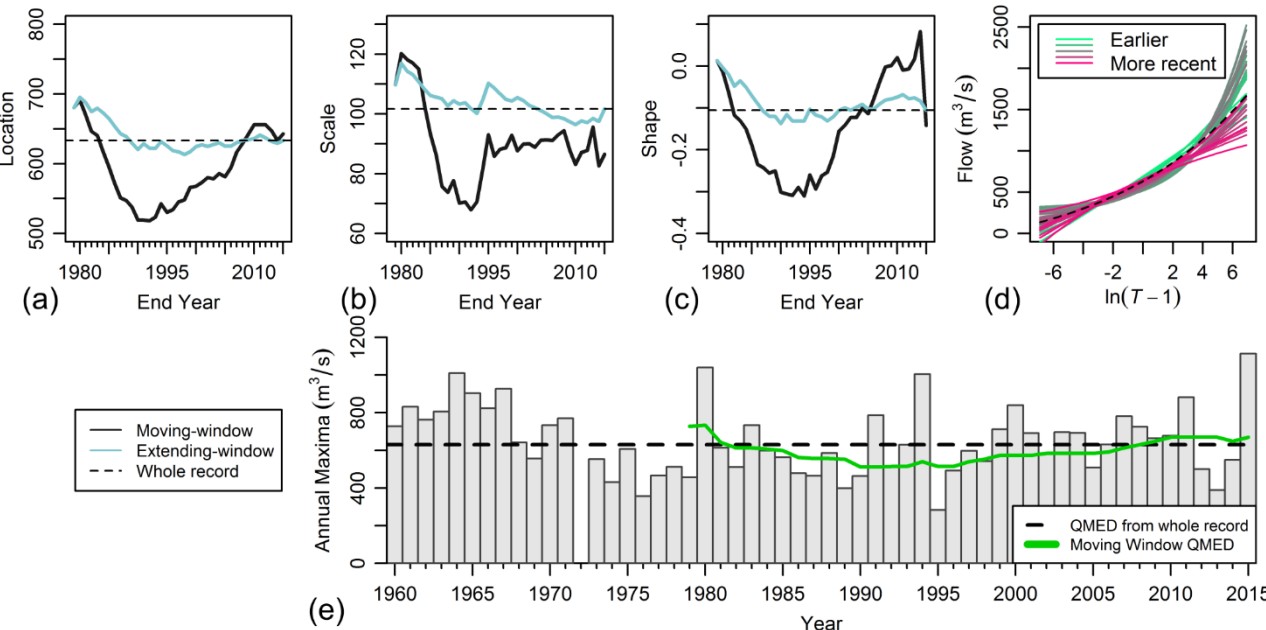

**Figure 3: Example of results from fitting stationary parameters on the Ribble at Samlesbury in different time windows. Parameters (a-c) computed under moving windows, extending windows and on the whole record. AMAX series with QMED moving estimate (e), and flood frequency curves generated from the moving window analysis and from the whole record (d). Note that the value from 1972-73 is missing, but this does not affect the analysis.**

## 3.2 Extending window analysis

An alternative approach to analysing change in flood regime is to adopt an extending window approach, which matches the standard practice of recomputing the flood frequency distribution upon acquiring new data. In the present study, the window initially includes the first 20 years of the record and is extended year-by-year to eventually cover the whole record. For example, a station for which records start in 1901 would be investigated using windows covering the years 1901-1920, 1901-1921, etc., up to 1901-2016. As before, stationary parameters are fitted and the value of QMED computed at each station using only the data inside the window. The purpose of using stretching windows is to see how specific events, once included, affect the values of the stationary parameters and return periods of large events.

Trends within extending windows start similarly to those within fixed-width windows, but gain an increasing amount of attenuation and lag, as older events never drop out of the window. This attenuation and lag means that the flood frequency curves developed for extending windows do not vary as much as for fixed-width windows. Use of an extending window can therefore mask periods of record during which the distribution of AMAX events can be quite different from the average, or mask changes in flow regime. However, extraordinary events do still have noticeable effects on the stationary location and shape parameters in particular.



### 3.2.1 Comparison of moving window and extending window analysis

A typical example of moving window estimates is presented for the Ribble at Samlesbury (NRFA station 71001) in Fig. 3. As a number of large events (bigger than QMED) "drop out of memory", $\xi$ decreases and $\kappa$ becomes more extreme, moving away from zero due to the difference between the smallest and largest events in the window. As the big events in 2000 and 2011

appear, the location parameter $\xi$ moves back the other way, whereas the large number of similar-sized events in this period lead to $\kappa$ moving towards and eventually passing zero, to become positive again. However, a very extreme event in 2015 leads to a massive shift in the shape parameter, which becomes more negative again.

These changes can be clearly seen in the flood frequency curves (Fig. 3(d)), where the curves from the middle period are more extreme due to more negative values of $\kappa$, but the later curves are more elevated around QMED, where the reduced variate

$\log(T-1)$ is close to zero, due to large values of $\xi$. This suggests that $Q_{100}$ and $Q_{50}$ estimates decrease towards the end of the record, but $Q_5$ and QMED are increasing towards the end of the record.

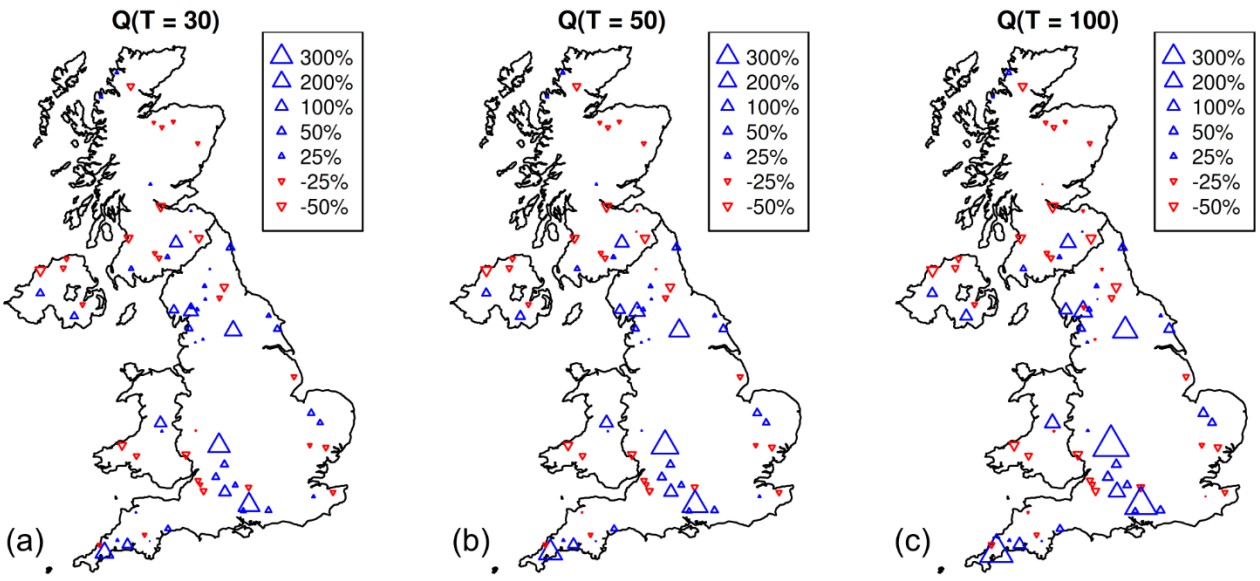

**Figure 4: Spatial distribution of trends in the 30-, 50- and 100-year return period floods across the UK Benchmark Network, comparing estimates (percentage increase) from the whole record to estimates just using the first 20 years of record at each station.**

For the extending window analysis, the lengthening record leads to more stable estimates over time. However variation in estimates can be seen throughout the record, suggesting a lack of convergence to a steady value, particularly in location and shape parameters. Single events such as the low value of 1995 have marked effects. The flood frequency curves under the extending window analysis (not shown) present a similar evolution in flood frequency to that of the moving window, but the curves on the whole show less inter-year variation. The extending window estimate for QMED is still fairly insensitive to the

extreme events (both large and small) as record lengthens, even less so than from the moving window; QMED is in many cases chosen over mean annual flood as a primary descriptive statistic for this insensitivity to single extreme events.





### 3.2.2 Spatial patterns of trends as records lengthen in the UK

To see the effects of the use of an extending window over the whole of the UK, Fig. 4 demonstrates the difference between using the start of the record and the whole record for the 30-, 50- and 100-year return period events, corresponding to the 3.33%, 2% and 1% AEP events under stationary conditions. Assuming they come from the same distribution, one might expect little variation between the two estimates (so the percentage difference plotted is close to zero). However, strong differences of up to 100% increase have been observed across England at all three return periods. Elsewhere the signal is less clear. Common patterns observed at all return periods are mostly down to the fixed expression for $Q_T$ conditional on the parameter values.

### 3.3 Non-stationary analysis

To look at how the stationary estimates compare to the non-stationary, parameters that vary linearly in time $(\xi(t), \alpha(t), \kappa(t))$ are fitted to the entire record at each station using maximum likelihood estimators.

### 3.3.1 Non-stationary Generalised Logistic distribution

To describe the changing distribution of the AMAX series over time, the stationary parameters are replaced by parameters that change linearly in time

$$\xi(t) = \xi_0 + \xi_1 t, \qquad \alpha(t) = \alpha_0 + \alpha_1 t, \qquad \kappa(t) = \kappa_0 + \kappa_1 t$$

where $t$ is the number of years since the start of the record. In order to fit these linearly varying parameters, maximum likelihood estimators (rather than L-moments) are determined on the AMAX series. Much work has been done investigating linearly changing location and scale parameters $(\xi, \alpha)$ for the Generalised Extreme Value distribution (GEV) distribution (Cunderlik and Burn, 2003; Leclerc and Ouarda, 2007; O'Brien and Burn, 2014). The shape parameter is typically left constant due to the high level of uncertainty in estimating the shape parameters even on long records (Coles, 2001). However to explore how these shape parameters might be changing in time and space, a linearly changing value of $\kappa$ is also included here. It should be noted, however, that since the shape parameter is bounded by -1 and 1, there is a limit to the length of period that the linear trend for $\kappa$ can be considered reasonable. Additionally, due to very different behaviours of the GLO for positive and negative values of $\kappa$, it is more physically realistic to expect a decay towards zero than a linear trend past zero.

### 3.3.2 Non-stationary return periods

The standard definition of the return period of flow $Q\ (T_Q)$ is intrinsically linked to the annual exceedance probability (AEP), the probability that a flow of given discharge $Q$ is met or exceeded within a given year. For example, the 1-in-100-year event has an AEP of 1%. However, when the probability of exceedance changes over time, due to the changing distribution, the notion of a return period should be updated similarly. Hu *et al.* (2017) focus on reliability of engineering structures, related to



the probability of failure over the design life of the structure. For example, if the design lifespan is $L$ years, then the survival probability of a structure built in year $y$ would be $P_{survival} = \prod_{s=y}^{y+L}(1 - P_Q(s))$. In this work, the return period must take into account the point of reference of interest, similar to the design life of a piece of hydraulic engineering like a dam or bridge. Using the definitions from Salas and Obeysekera (2014), the return period of an event with flow exceeding $Q$, starting from

year $y$ is given by

$$T_Q(y) = \sum_{r=y}^{\infty} \prod_{s=y}^{r} (1 - P_Q(s))$$

where $P_Q(s)$ is the annual exceedance probability of a flow $Q$ in year $s$. If the probability of exceedance is the same for each year (stationary), this can be simplified to give

$$T_Q = \frac{1}{1 - P_Q}$$

which matches with the standard conversion from AEP to return period (Hosking and Wallis, 1997).

The non-stationary estimate for the T-year flood, starting from year y, $\widetilde{Q_T}(y)$, is obtained by inverting $T_Q(y)$. However, this is done numerically due to the intractability of the expressions involved. It should be observed that if $P_Q(s)$ decreases sufficiently quickly, it is possible for the value of $T_Q(y)$ to be infinite. This might be the case where an observed upper bound of flood magnitudes decreases over time, such that a value of interest $Q*$ goes from below to above the upper bound (Salas and

Obeysekera, 2014). In cases like this, a flood of magnitude $Q*$ will never happen again, unless the trend or distribution changes.

### 3.3.3 Results based on whole record

Across the 73 study catchments, the different types of trend can be divided according to the direction of movement in the median (QMED or $\xi$ increasing or decreasing) and extremes of the flood frequency curve ($\kappa$ tending towards and away from zero). In some cases, a parameter may reverse its direction of travel one or more times as the window is moved from the start

to the end of the record, resulting in a flood frequency curve with an inconsistent time-dependency.

Fig. 5 shows the size and direction of the parameters $\xi_1$, $\alpha_1$, $\kappa_1$ fitted to each of the 73 full AMAX records (i.e. the year-on-year change). The relative changes of location and scale parameters are shown ($\xi_1/\xi_0$, $\alpha_1/\alpha_0$), while for $\kappa_1$ the actual value is shown. Figure 5 confirms that positive trends in the estimates for the location parameter $\xi$ are more numerous and typically larger than negative trends (56 positive vs 17 negative), and shows that the largest positive trends cluster around the England-

Scotland border, where $\xi$ can increase by 1-2% per year.



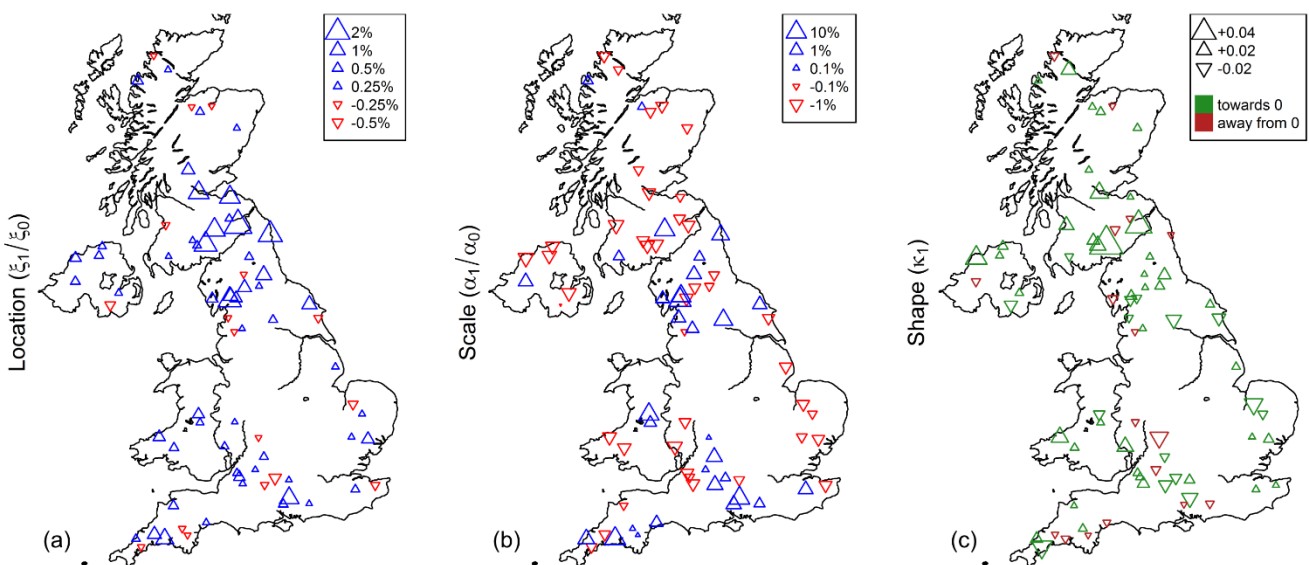

**Figure 5: Maps of UK Benchmark Network showing relative spatial trends in $\xi_1/\xi_0$ (a), $\alpha_1/\alpha_0$ (b), and $\kappa_1$ (c).**

For the scale parameter $\alpha$, there is less spatial consistency in the size and direction of trends, with 42 negative and 31 positive values of $\alpha_1$. However, there are no extreme negative trends ($\alpha_1/\alpha_0 < 0.02$), while 11 positive trends are greater than 2%, the

most extreme case, $\alpha_1/\alpha_0 = 0.167$, implies that $\alpha(t) = \alpha_0 + \alpha_1 t$ reaches double $\alpha_0$ after approximately six years. The shape parameter $\alpha$ has the greatest influence over the gradient of the flood frequency curve in the centre of the distribution away from the tails, so increased values of $\alpha$ suggests increases in the ratio between magnitudes of more frequent floods (2-year-flood and 30-year-flood, for example). It should be noted to obtain estimates with the same level of uncertainty, a longer AMAX series is required for $\alpha_1$ than for $\xi_1$.

While the magnitudes of $\kappa_1$ are well balanced either side of zero, trends towards zero are more numerous than trends away from zero (56 towards vs 17 away). Smaller values of $\kappa$ (i.e. closer to zero) have the effect of straightening the flood frequency curve (when return periods are plotted as their logistic reduced variate), which in cases with no upper bound (most cases) has the effect of reducing the ratios between extreme events (e.g. between the 1% AEP and 3.33% AEP floods for a fixed year). Even more so than for the scale parameter, it should be noted that an even longer AMAX series is required for a specific level

of uncertainty in $\kappa_1$ than for $\alpha_1$ or $\xi_1$ and that this has been given as a reason in previous studies not to quantify trends in $\kappa$ (O'Brien and Burn, 2014).

For most records, the overall trend is towards an increase in $\xi$, corresponding to an increase in QMED and other large floods. However, most stations show a trend in $\kappa$ whereby its value moves closer to zero. This trend exists for both $\kappa$ negative and increasing and $\kappa$ positive and decreasing, and has the effect of "straightening" the flood frequency curve: this reduces the ratio

between magnitudes of extreme events (e.g. the 1% AEP and 0.1% AEP events for a given year) in cases where $\kappa$ is negative and increasing.



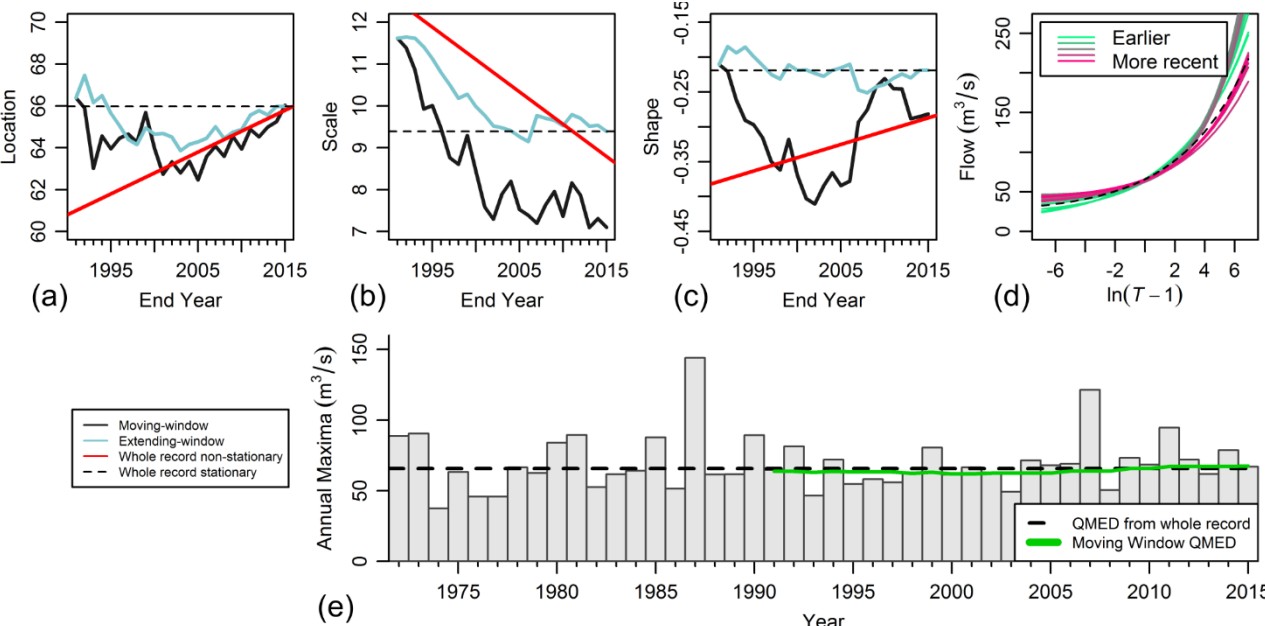

**Figure 6: Example of results from fitting stationary and non-stationary parameters on the Agivey at Whitehill in different time windows. (a-c) Stationary parameters computed under moving windows, extending windows, and the whole record, and non-stationary parameters computed on the whole record. (e) AMAX series with QMED moving estimate. (d) Flood frequency curves generated from the moving window analysis and generated from the whole record.**

Although non-stationary parameter fitting over the whole record allows trends to be quantified and compared easily, it can only register the cumulative total trend even if a trend changes direction several times over the period of record. An example of this is demonstrated at Agivey at Whitehill (NRFA station 203028, Fig. 6), where, for a 20-year moving window, κ falls from −0.21 to −0.40 before increasing again to −0.28, corresponding to the occurrence of the largest event in 1987, with the only other similarly large event in 2008. In contrast, for non-stationary parameter fitting, $\kappa$ increases linearly by 0.038 per year. Although the changes are not consistent over time, a flattening of the flood frequency curve can still be observed, along with a decrease in $Q_{100}$ (reduced variate of 4.59) despite a steady value of QMED.

Non-stationary parameter estimates can highlight issues with using a single value to represent parameters over a changing catchment, but can also illustrate that stationary parameter estimates are not necessarily an average of the non-stationary parameter estimates. Figure 7 highlights one such example from Gifford Water at Lennoxlove. Under the stationary distribution, the shape parameter estimate is approximately $\kappa$ = -0.2, whereas $\kappa(t) \approx -0.45 + 0.001t$ is the non-stationary estimate, which is quite different. This can be explained by considering the fixed window at the start of the time series and the end of the time series. By recalculating the plotting positions for these windows, the curves using the non-stationary estimates for the first and last year of records describe the early and late years well, respectively. On the other hand, the stationary estimates fit the whole curve well but not any one section, as the very different behaviours at each end "average" out.





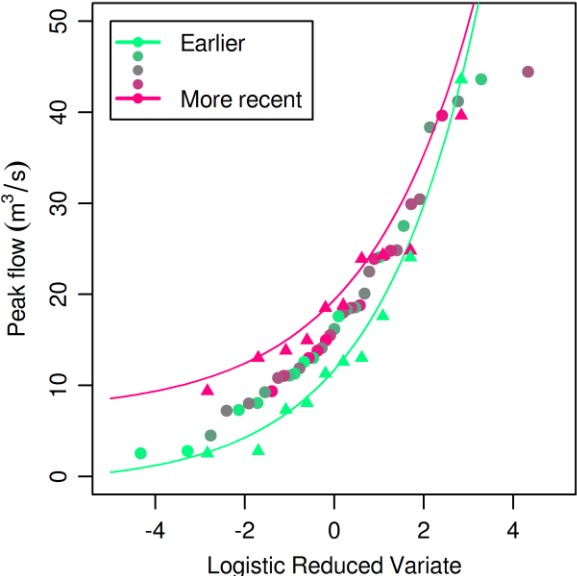

**Figure 7: Flood frequency curves (FFCs) for Gifford Water at Lennoxlove. i) AMAX plotted for first ten years of record (green triangles) and FFC plotted using $\xi(t), \alpha(t), \kappa(t)$ from first year of record (green line); ii) AMAX plotted for last ten years of record (pink triangles) with FFC plotted using $\xi(t), \alpha(t), \kappa(t)$ from last year of record (pink line); iii) AMAX plotted for whole record (circles). Each set of AMAX points plotted using separately calculated plotting positions.**

Although this work does not attempt to attribute causes to the trends in UK Peak Flow data, it is of interest to see whether any standard covariates correlate strongly with the trends observed. Figure 8 shows relative change in $\xi$ ($\xi_1/\xi_0$), relative change in $\alpha$ ($\alpha_1/\alpha_0$) and absolute change in $\kappa$ ($\kappa_1$) against catchment centroid easting, catchment centroid northing, average annual rainfall during 1961-1990 (SAAR) and catchment area. This reveals few strong relationships between trends in any GLO parameter and either catchment location, size or yearly rainfall. The most obvious thing observed is that there are no strong trends in $\alpha$ for catchments larger than around 700 km$^2$ or for the most northerly catchments in the UK. In practical terms, this means that the gradient of the centre of the flood frequency curve is relatively unchanging over time for catchments in Scotland and the larger catchments elsewhere in the UK.

### 3.4 Non-stationary extending window analysis

One can also investigate the sensitivity of non-stationary parameters to new data. Starting with the record up to 2000, the non-stationary parameters were refitted for each station after adding one new year at a time. In general, $\xi_0$ and $\xi_1$ changed in opposite directions due to the fact that QMED derived from the whole record (which roughly associates to the average of $\xi(t)$ over the record) varies slowly, but the addition of an extreme point often changes the slope of the linear fit, so $\xi_0$ has to change conversely to compensate. This is less marked in $\alpha(t)$. In cases where there is low variability in AMAX, the values of $\kappa_1, \alpha_1$ and $\xi_1$ vary slowly, but in many cases these parameters vary erratically.





**Figure 8: Scatterplots plotting catchment descriptors against relative trends in non-stationary parameters.**




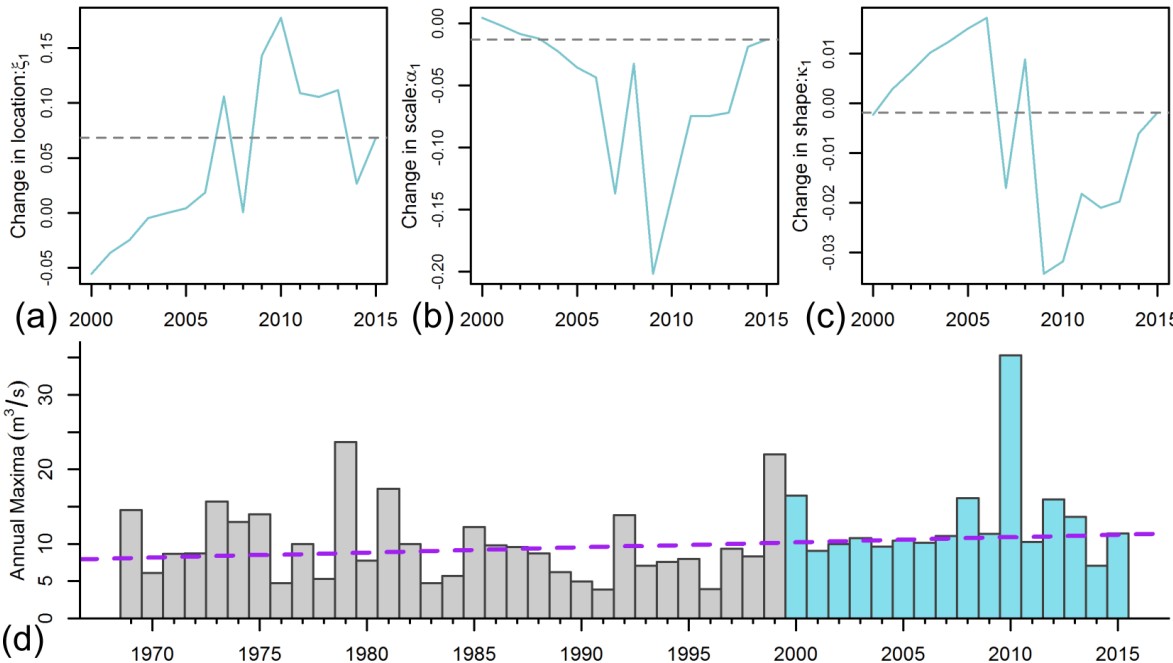

**Figure 9: (a-c) Non-stationary parameters using fixed-start extending windows on the Warleggan at Trengoffe showing change due to addition of single data points, $\xi_1$, $\alpha_1$ and $\kappa_1$ plotted. (d) AMAX series highlighting period of extending windows.**

Figure 9 shows an example where the presence of an extreme event massively changes the trend parameters. The event in 2010, which far exceeds any previous event, has a very large effect on $\kappa_1$ and $\alpha_1$, causing both to become significantly more negative. Compare this to the relatively smooth variation before 2006 which could be linked to the very consistent values of AMAX close to QMED in 2000-2006, which would lead to an increase in $\kappa(t)$ and drop in $\alpha(t)$, and push $\xi_1$ closer to zero. It should be noted, though, that this example is extremely clear, most stations showed these kinds of effects but with much more variability.

## 3.5 Changes in flood return periods

Finally the 30-year and 50-year floods are compared for each of the stations with records extending up to at least 2015, under stationary and non-stationary estimates. The value of $Q_{30}$ and $Q_{50}$ are computed from the stationary parameter estimates. Then, using the non-stationary parameter estimates, the annual probabilities of exceedance $P_Q(y)$ are computed, assuming that the fitted non-stationary parameters remain valid for the 50 years following the start of the record at the station (66 stations satisfied this). These are used to find the return period function $T_Q$ as in Section 3.3.2. $\widetilde{Q_{30}}(y_0)$ and $\widetilde{Q_{50}}(y_0)$, with $y_0$ equal to the start of each station's record, are then computed by inverting the function. Note that, for numerical tractability, the sum was truncated once the value of summands became less than 0.01. The values obtained were tested and seen to be fairly insensitive to the exact threshold for truncation, as long as it was sufficiently small (much less than 1).





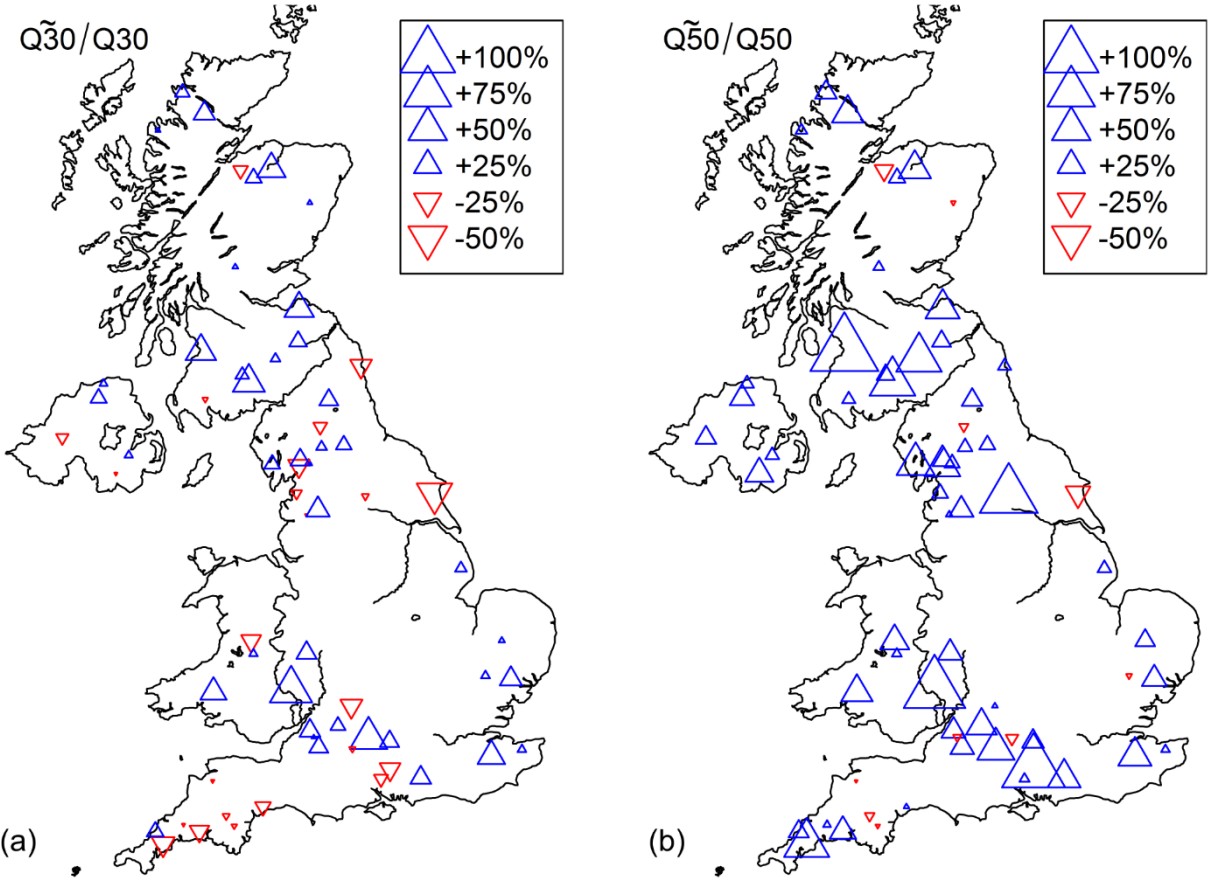

**Figure 10: Percentage change of long-return period flood magnitudes** $\left(\left(\frac{\widetilde{Q_T}}{Q_T} - 1\right) \times 100\%\right)$ **computed under assumptions of stationarity ($Q_T$) and non-stationary probability of exceedance ($\widetilde{Q_T}$). Shown for the 30-year event (a) and 50-year event (b).**

Figure 10 shows the ratios of $\widetilde{Q_{30}}(y_0)/Q_{30}$ and $\widetilde{Q_{50}}(y_0)/Q_{50}$. Note that if these changes are greater than zero, then this suggests

5    that the event under non-stationary assumptions is larger. In general, we see a quite mixed signal for the 30-year event, suggesting that many of these estimates are quite similar under stationary and non-stationary assumptions. However, for the 50-year event we see a more consistent increase in magnitude compared to estimates made under stationary assumptions at the start of the period of record. This is likely due to the continued increase in location and scale parameters over time (compare to Fig. 5), causing an ever increasing discrepancy between the stationary and non-stationary probabilities of exceedance. As

10    in Fig. 5, the biggest changes are on the England-Scotland border. Due to the limitations of the linear expressions used for the non-stationary parameters, it was not possible to estimate the non-stationary 100-year flood at many of the locations in the UKBN2. More flexible expressions for the parameters, $\kappa(t)$ in particular, may improve this.





## 4 Conclusions

In this paper, the updated UK Benchmark Network has been used as a near-natural set of example stations to investigate how the inclusion of new data affects flood frequency estimates under both stationary and non-stationary assumptions. The change in median behaviour by the addition of larger and smaller events was reaffirmed, but the big changes in the shape parameter $\kappa$

due to the influence of extreme events, leading to much steeper flood frequency curves in the upper tail, have also been presented. In addition to observation of new data, this could reaffirm the notions of "in living memory" as unreliable since, as small events are forgotten, the relative sizes of the more recent floods may be distorted. The fixed-width moving window analysis can be seen as a proxy for this.

To put this in context in the UK, Fig. 5 suggests flood frequency curves are flattening, suggesting that the most extreme floods

may not necessarily be getting bigger, but that the more likely floods, such as the 20% AEP period flood, may be getting larger, a similar conclusion to that of Hirsch and Archfield (2015). In some locations such as Southern Scotland, patterns suggest a reduction in QMED and short-return period floods. The effect of adding data to the AMAX series in the context of non-stationary estimates was also investigated. It showed that the addition of single events was enough to have a marked difference in the non-stationary parameter estimates, which in turn can have a big impact on the estimates of, for example, $Q_{50}$ and $Q_{100}$.

This, along with the fact that the empirical plotting positions of very extreme events may massively over-estimate their frequency, means that a single large event should be considered within the framework of the underlying hydrological processes. Finally, the concept of the return period was discussed, with the non-stationary return period using a time-varying probability of exceedance based on non-stationary parameter estimates of the Generalised Logistic distribution.

This study has shown that the difference between using return periods based on stationary distributions and non-stationary

distributions can be significant, such that the "70-year design life" of a structure built 30 years ago may be inaccurate to the point of being unfit for purpose. However, as discussed above, the introduction of new data can vastly change estimates if the new data are extreme. In this case, one needs to examine new data and its effect on current estimates to determine whether the change is reasonable. If several new data points are obtained which suggest a different model, then the new data can be more reasonably included. On the other hand, the fact still remains that, as seen above, increased volumes of data allow for reduced

uncertainty and hence one should not exclude old data without good reason.

In the future, the use of fixed-width moving windows would be very valuable in the study of flood-rich/flood-poor period quantification in river flow data. If these periods can be elucidated, it would be of interest to examine the underlying hydrological mechanisms. On a shorter timescale, the moving window approach could offer some insight into seasonality modelling in flood frequency estimation.

**Author contributions**

ES coordinated the study, helping write the manuscript and added context to the research. GV generated all the figures and helped write the manuscript. AG designed and performed the analysis and helped write the manuscript.



**Competing interests**

The authors declare they have no conflict of interest.

**Acknowledgements**

The authors would like to thank Giuseppe Formetta and Chingka Kalai, and to thank the editors and reviewers for their
invaluable feedback. This work was funded by the Centre for Ecology & Hydrology through the FEH Research Programme
within the Hydro-climatic Risks science area.

**Data availability**

The data was all obtained freely from the National River Flow Archive (Version 6: 2017).

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
