# Peer review of "Have trends changed over time? A study of UK peak flow data and sensitivity to observation period."

_Natural Hazards and Earth System Sciences, 2019_

## Referee Comment (RC1) · Anonymous Referee #1 · 8 May 2019

The paper "Have trends changed over time? A study of UK peak flow data and sensitivity to observation period" by Griffin, Vesuviano and Stewart presents an investigation of changes in the parameter estimates of the GLO distribution through time for British network of near natural catchments.

The paper introduces some interesting approaches to quantify and visualise changes trough time. It reads well, is well organised and has suitable tables, figures and references. I feel there is maybe not a very clear focus in the results presented by the authors: the study is interesting and well executed, but there isn't a final clear concept that emerges as the final take home message in the paper other than "it's compli-

cated" (which is a good take home message but one that was already known before this paper). From a more technical point of view I think the authors lack a discussion of uncertainties in the estimation (see more on this below) and of the possible correlation/interaction between the parameter estimates, either for the stationary of non-stationary case. As they also state in the end of the paper the final estimates obtained for the design events of interest depend on the estimated values of all parameters, so that even if the shape parameter is estimated to be closer to 0 (rather than negative) the final estimates of the 50-year are still larger, because of the changes in the location and scale parameter. This can be difficult to understand and accommodate, and I think it would deserve a larger exploration and discussion in the paper.

The other point I think the authors need to reflect on is the choice of link functions used to model the distribution parameters: I believe that linear trends might not be the most suitable ones for this application (again see more on this below).

**Some other specific comments**

Page 3 - line 4: an initial version *of the benchmark network* (to clarify it is not the NRFA the authors are talking about).

Page 3 - line 9: from the writing I understand the data used is the instantaneous peak flow data - not the daily. Maybe this could be specified more clearly since the proportion of daily data missing is mentioned above.

Page 3 - line 13-14: I imagine this is because this is the area of the country with the most urbanisation, but this could be spelled out for those not familiar with British geography.

Page 3 - line 26: this is a very good point, often overlooked in practice. In the FEH estimation procedure though xi and QMED are constrained to be the same I recall - but I gather the authors do not attempt to do that in this paper.

[Figure]

In Figure 3 in the extended window there seems to be some correlation between the functional shapes of the scale and the shape parameters. From experience of fitting extreme value distributions to at-site data I know that especially when large events are added to the analysis dramatic changes in the shape parameter are sometimes also connected to fairly large reductions in the scale parameter: this makes sense as some of the variation in the data is now explained by a higher skewness instead of a large variability. I wonder if the authors could comment on this and if they have noticed a similar phenomenon in their moving averages.

Section 3.2.1/Figure 3 - since you use the greek letters to discuss the values of the parameters I would add them to the plots so it is easier for the reader to connect the text and the figure. Alternatively you could use the words location. scale, shape in the text.

Page 8 - line 7: would it be the case that opposite signs could be seen for the 2-years and 5-years events as in the case study presented in Figure 3?

Section 3.3.1: are the three linear models fitted separately or is this one unique linear model fitted to all the AMAX (in which case I am impressed things converge with no problems). Also, maximum likelihood is used in the estimation changing the estimation procedure, maybe using L-moments for trends as in Jones (2013) could have been relevant in this context. It is a bit odd that two estimation approaches are used to find trends, ML could have been easily employed to do the moving averages as well (probably leading to very similar results). On the other hand using ML for the moving average would have possibly allowed the estimation of some form of uncertainty, to assess whether the apparent shifts in the parameter values are not contained within the sampling variability. In general uncertainty/variability in the estimation is not mentioned at all in the paper, while it could well be that the changes in the point estimated identified by the authors are swamped by the variability of the estimation.

Page 8: line 22. The authors discuss some issues connected to the fact that the linear

form imposed to the shape parameter means one should be careful when extrapolating outside the time range used in the regression. Note that this is also technically true for the scale parameter as well, which should be positive. Later in the paper the authors point out that that the linear form used for the shape also makes it impossible for them to calculate some of the percentage changes. I would imagine that using some form of truncated logistic regression or some other link function in the model (see the mgcv::gevlss function in R) would make fix some of these problems? I understand this would require the complete reworking of the findings - but it would seem the reasonable thing to do.

Page 9 - line 2: what is $P_Q(s)$ ? I see it is defined later - maybe this paragraph could be rewritten to make this clearer

Page 10, line 4: why is 0.02 an extreme negative trend? (I mean if you miss a -, and I am not clear if 0.02 would be linked to some specific large change in the design event).

Page 10, figure 5, right panel: red and green are the definition of things colour blind people can not distinguish, maybe use purple and yellow?

Page 11 - line 16: "which is quite different" in what sense? Maybe useful to give the range of the values (i.e. what is the maximum of it) or to comment more on what you mean by quite different. I also think this has something to do with the fact the location and scale parameters are also estimated to span quite different values in the non-stationary model than in the stationary model. Finally as mentioned before: is this difference significant?

Page 14 - section 3.5: I am not entirely clear on what is being described here. Why does the assumption that the non-stationary parameters are valid for more than 50 years only hold for 66 stations? Are these stations with more than 50 years or stations for which the $\kappa(t)$ function stays within the required bounds? Do I understand correctly that you are using L=50 and applying the formulae shown in Section 3.3.2.

[Figure]

References

D. A. Jones; On an extension of the L-moment approach to modelling distributions which include trend. Hydrology Research 1 August 2013; 44 (4): 571–582. doi:

---

## Referee Comment (RC2) · Anonymous Referee #2 · 21 Jun 2019

The paper "Have trends changed over time? A study of UK peak flow data and sensitivity to observation period" by A. Griffin et al. analyses the changes in time of the parameter estimates of the Generalized Logistic distribution and flood quantiles for the flood data of the UK Benchmark Network. The authors use two approaches (i.e. fix-width moving window and fixes-start extending window) to investigate the sensitivity of the parameter estimates to record length and to the presence of most extreme events, under both stationarity and non-stationarity assumptions.

The manuscript is well written, the aim of the paper is clearly stated in the introduction and the analyses and the results are presented in an appropriate way. The methods/approaches are not particularly new, but the results (especially the maps showing the spatial distribution of the trends in the quantiles and parameters) are of clear scientific and technical interest, given that the detection of flood regime changes is a topic of major concern and relevance.

I would nevertheless suggest to the authors some changes concerning mainly the text and the organization of the paper in the result and conclusion section:

Page 1 – Lines 10-11: from this sentence in the abstract it seems that the aim of the paper is to separate the effects of land-use change from climate change. The UK Benchmark Network is used instead to consider near natural catchments only. I would suggest to the authors to rephrase this sentence.

Page 1 – Line 29: please define NRFA in the text, I see it is defined at page 3 but you mention it two times before in the text.

Page 2 - Lines 8-9: I would clarify in the text that Hall et al. (2014) is a review article. The same at page 4 – lines 8.

Page 2 - Line 30: I haven't fully understood the third listed objective. In my opinion it is unnecessary.

Page 3 - Line 11: please define AMAX in the text

Page 3 - Line 26: please put numbers to the equations

Page 3 - Line 30: I understand the meaning of the sentence but, to be precise, it is not correct to say that T is equivalent to the annual exceedance probability, but rather that they have a one-to-one correspondence according to the given relationship.

Page 4 - Figure 1: It would be helpful to add to this map the locations and the names of the hydrometric stations that are taken as examples later in the manuscript (i.e. the stations of figure 3, 6, 7 and 9). In this way the reader would be able to find also in the maps (e.g. in figure 4) what is discussed at the level of the single station. I would also

suggest using different (maybe solid) colors because I find the map not easy to read (the blue and green are very similar).

Page 4 - Line 9: The authors state in section 2.1 that the minimum record length is 21 years therefore isn't this sentence unnecessary?

In section 3.1 and (beginning of) section 3.2 of the results (page 4, 5 and 6) the authors mainly describe the moving and extending window approaches, making general considerations and without directly mentioning the results of the study nor the figures. I would suggest revising the organization of these sections (for example by moving the parts that are descriptive of the approaches into the method section) or to refer directly to the figures and results, while describing the analysis. The same applies to section 3.3.1 and 3.3.2 where the authors give definitions of the non-stationary parameters and return periods.

Page 6 - Figure 3: Why don't the authors plot also the line corresponding to the extending-window in panel e (which is instead mentioned at page 7 – lines 19-20)? I would also mention somewhere in the figure caption that the parameter and Q_MED values are plotted in correspondence of the end year of the moving and extending windows.

Page 8 - line 16-17: Why do the authors use different methods for parameter estimation in the stationary and non-stationary case? Please provide some explanation for this choice or use the same method for both.

Page 8 - line 19-24: The authors use a linear regression with time for the shape parameter, but a convincing justification for this choice is not given; they highlight instead its negative implications and limitations (also at page 15 – lines 10-12). Please provide some explanation for the choice of this relationship. In agreement with the comment of the Anonymous Referee #1, I believe it's reasonable to try another expression for k(t) that overcomes the current limitations.

[Figure]

Page 9 – line 6: I was not able to find this exact formulation of the return period in Salas and Obeysekera (2014). Is there an assumption about the condition of non-stationarity (increasing, decreasing or shifting extreme events), as done in Salas and Obeysekera (2014)? Can the authors comment a bit more on this definition?

Page 9 – line 9: If $P_Q$ is the annual exceedance probability, as defined at line7, I believe there is a typing error in the equation. Shouldn't $T_Q$ be equal to $1/P_Q$?

Page 11 – lines 19-20: The authors talk about figure 7 and refer to the stationary estimates that are not shown there. Please add them.

Page 12 – lines 6-13: I find figure 8 interesting, but I think its description in these lines is a bit synthetic and could be improved (only 2 panels out of 12 are actually commented).

Page 16 - lines 11-12: I believe that this statement about $Q_{MED}$ is in contrast to what observed in figure 5, panel a, and what stated at page 9 – lines 23-25.

Section 4: In my opinion the organization of this section can be improved; it is a bit confused at the moment and, as the Anonymous Referee #1 also comments, there is no clear conclusion or take-home-message emerging. I would be appropriate to refer to the initial objectives, stated at page 2 – lines 27-30, and to re-organize this section accordingly, in order to clearly demonstrate how the analyses in the paper have fulfilled the initial objectives.

---

## Author Comment (AC1) · 5 Aug 2019

**NOTE:** **Referee's comments in black Segoe UI,**

**Our responses in blue Times New Roman.**

**Our references to page and line numbers are relative to the discussion paper, not to the revised manuscript.**

**Anonymous Referee #1**

The paper "Have trends changed over time? A study of UK peak flow data and sensitivity to observation period" by Griffin, Vesuviano and Stewart presents an investigation of changes in the parameter estimates of the GLO distribution through time for British network of near natural catchments.

The paper introduces some interesting approaches to quantify and visualise changes trough time. It reads well, is well organised and has suitable tables, figures and references. I feel there is maybe not a very clear focus in the results presented by the authors: the study is interesting and well executed, but there isn't a final clear concept that emerges as the final take home message in the paper other than "it's complicated" (which is a good take home message but one that was already known before this paper). From a more technical point of view I think the authors lack a discussion of uncertainties in the estimation (see more on this below) and of the possible correlation/interaction between the parameter estimates, either for the stationary of non-stationary case. As they also state in the end of the paper the final estimates obtained for the design events of interest depend on the estimated values of all parameters, so that even if the shape parameter is estimated to be closer to 0 (rather than negative) the final estimates of the 50-year are still larger, because of the changes in the location and scale parameter. This can be difficult to understand and accommodate, and I think it would deserve a larger exploration and discussion in the paper.

The other point I think the authors need to reflect on is the choice of link functions used to model the distribution parameters: I believe that linear trends might not be the most suitable ones for this application (again see more on this below)

The authors thank the referee for these comments, and hope that they can act on them to the satisfaction of all concerned. Primarily, this has involved re-performing the analysis using different link functions for the scale and shape parameters, and restructuring the methods and conclusion. This has resulted in the regeneration of several figures, but they do not greatly change

the conclusions we make, but a change was noted and so comments about Figure 5 have changed to deal with this (see specific comments below). We also expand on the observed increase in $Q_{50}$ (compared to a stationary estimate) despite the shape parameter moving towards zero when relevant around figures 4, 5, 8 and 10.

**Some other specific comments**

Page 3 - line 4: an initial version *of the benchmark network* (to clarify it is not the NRFA the authors are talking about).

Noted. The suggested four words of text have been added to this sentence.

Page 3 - line 9: from the writing I understand the data used is the instantaneous peak flow data - not the daily. Maybe this could be specified more clearly since the proportion of daily data missing is mentioned above.

To address this, the suggested addition of "instantaneous annual maximum" has been added to line 14. On line 9, "1.5% of missing daily data" has been replaced by "1.5% of days missing data".

Page 3 - line 13-14: I imagine this is because this is the area of the country with the most urbanisation, but this could be spelled out for those not familiar with British geography.

This is a point we agree is worth making. To rectify this, "more heavily urbanised" has been inserted into the sentence on page 3, line 13. It now reads: "Due to the requirement of UKBN2 that catchments must be free of significant land use change over the period of record, catchments in the more heavily urbanised south-east and midlands of England are fewer in number and typically smaller than catchments located elsewhere."

Page 3 - line 26: this is a very good point, often overlooked in practice. In the FEH estimation procedure though xi and QMED are constrained to be the same I recall - but gather the authors do not attempt to do that in this paper.

You are correct. Two sentences have been added after, specifically: "The FEH statistical method constrains QMED and $\xi$ as equal. However, this study does not."

In Figure 3 in the extended window there seems to be some correlation between the functional shapes of the scale and the shape parameters. From experience of fitting extreme value distributions to at-site data I know that especially when large events are added to the analysis dramatic changes in the shape parameter are sometimes also connected to fairly large reductions in the scale parameter: this makes sense as some of the variation in the data is now explained by a higher skewness instead of a large variability. I wonder if the authors could comment on this and if they have noticed a similar phenomenon in their moving averages.

This point is an interesting one. No clear correlation between shape and scale was observed across the dataset in our investigation. To avoid further complicating this paper with a full discussion this additional issue, we will not discuss this further in the paper. However, in the conclusions, the following sentence is added at page 22, line 24. "As mentioned in Section 3.5, an increase in long return period events is observed despite a drop in the shape parameter towards

zero. This is due to the overall increase in location and scale parameters over time. In some cases, correlation between change in scale and change in shape could be linked to the addition of a single large event. A full analysis of this correlation would be worthy of future investigation."

We additionally noted that including non-stationarity seemed to perform in the opposite direction, variation in the data being explained by changes over time rather than high skewness or variability. This is mentioned in the Conclusions as "On the whole, the inclusion of non-stationarity allows, in some sense, the ability to exchange "variability" (which leads to more extreme scale and shape parameters) for "change over time"".

Section 3.2.1/Figure 3 - since you use the Greek letters to discuss the values of the parameters I would add them to the plots so it is easier for the reader to connect the text and the figure. Alternatively you could use the words location, scale, shape in the text.

Agreed. Greek letters have been added to Figures 3 and 6, without removing the words "location", "scale", or "shape".

Page 8 - line 7: would it be the case that opposite signs could be seen for the 2-years and 5-years events as in the case study presented in Figure 3?

This is a good point. To rectify this omission, the following is changed on page 8 line 7: "Common patterns observed at all long return periods are mostly down to the fixed expression for $Q_T$ conditional on the parameter values. However, it is seen to be the case that opposite directions of change can be observed at some stations for the 2-year and 5-year events (e.g. Fig. 3)."

Section 3.3.1: are the three linear models fitted separately or is this one unique linear model fitted to all the AMAX (in which case I am impressed things converge with no problems). Also, maximum likelihood is used in the estimation changing the estimation procedure, maybe using L-moments for trends as in Jones (2013) could have been relevant in this context. It is a bit odd that two estimation approaches are used to find trends, ML could have been easily employed to do the moving averages as well(probably leading to very similar results). On the other hand using ML for the moving average would have possibly allowed the estimation of some form of uncertainty, to assess whether the apparent shifts in the parameter values are not contained within the sampling variability. In general uncertainty/variability in the estimation is not mentioned at all in the paper, while it could well be that the changes in the point estimated identified by the authors are swamped by the variability of the estimation.

The parameters are all fitted in a single model (which is nonlinear). Initially, the L-moments method was chosen to compare currently recommended FEH practice for flood frequency estimates to a non-stationary model. However, following this comment, we have rerun the analyses to use maximum likelihood in all cases. This has led to very little change in figures 3, 4, and 6, but all have been replaced with updated versions. Page 3 line 18 is changed to "To this end, the Generalised Logistic Distribution (GLO) is fitted using Maximum Likelihood estimators to the AMAX series…" Page 8 line 16 has had "(rather than L-moments)" removed. We felt Jones (2013) was an argument against using L-moments in this non-stationary setting.

Work concerning estimation of the sampling variability around a detected/fitted trend is already underway. The authors plan to write this up as a future journal article in due course.

Page 8: line 22. The authors discuss some issues connected to the fact that the linear form imposed to the shape parameter means one should be careful when extrapolating outside the time range used in the regression. Note that this is also technically true for the scale parameter as well, which should be positive. Later in the paper the authors point out that that the linear form used for the shape also makes it impossible for them to calculate some of the percentage changes. I would imagine that using some form of truncated logistic regression or some other link function in the model (see the mgcv::gevlss function in R) would make fix some of these problems? I understand this would require the complete reworking of the findings - but it would seem the reasonable thing to do.

We have repeated the analysis with new forms specified for both the shape and scale parameters as follows:

$\alpha(t) = \exp(\alpha_0 + \alpha_1 t)$

$\kappa(t) = 1.5 / (1 + \exp(\kappa_0 + \kappa_1 t)) - 0.75$ (equations replaced on page 8, line 15)

Unfortunately, the use of the GLO distribution restricts us from using *gevlss*, but the authors appreciate this recommendation.

In the body of the text, "linearly" is deleted from page 8 line 21 replaced by "based on the logit function". The sentences on page 8, lines 21-24 are replaced by:

"It should be noted, however, that the chosen form of $\kappa(t)$ means that the parameter value will tend towards +0.75 or -0.75 as $t$ approaches infinity, potentially passing through zero. Due to very different behaviours of the GLO for positive and negative values of $\kappa$, it is more physically realistic to expect a decay towards zero than a trend crossing zero".

Other references to "linear" are removed where appropriate, most notably in the final two sentences before the Conclusions section.

Due to this new investigation and the new function to describe the change in shape parameter over time, it was found that there was no example station in the present dataset which could be chosen to clearly describe the arguments referring to figure 7 (page 11, lines 15-20). Rather than give a poorly justified discussion or generate a synthetic dataset to explain this phenomenon, this section has been removed.

Page 9 - line 2: what is $P_Q(s)$? I see it is defined later - maybe this paragraph could be rewritten to make this clearer

The phrase "where $P_Q(s)$ is the annual exceedance probability of a flow $Q$ in year $s$" has been moved to immediately after the first occurrence of $P_Q(s)$ i.e. where $P_{survival}$ is defined (page 9 line 2).

Page 10, line 4: why is 0.02 an extreme negative trend? (I mean if you miss a -, and I am not clear if 0.02 would be linked to some specific large change in the design event).

This is a good point. Since the figure was redrawn using the new analysis, the paragraph has been updated and updated. The sentences at the top of page 10 now read: "For the scale parameter $\alpha$, there is less spatial consistency in the size and direction of trends, with 37 positive and 36

negative values of α1. Four positive and no negative trends are greater than 2%, the most extreme case, α1 = 0.030, implies that α(t) = exp(α0 + α1t) doubles every 23.2 years."

Page 10, figure 5, right panel: red and green are the definition of things colour blind people cannot distinguish, maybe use purple and yellow?

Figure 5 right panel has been re-drawn using purple and green. In addition, figures 1, 2, 3, and 6 have been re-drawn using Viridis colour scale D, which is colour-blind and monochrome friendly.

[Figure]

Page 11 - line 16: "which is quite different" in what sense? Maybe useful to give the range of the values (i.e. what is the maximum of it) or to comment more on what you mean by quite different. I also think this has something to do with the fact the location and scale parameters are also estimated to span quite different values in the non-stationary model than in the stationary model. Finally as mentioned before: is this difference significant?

Since, as mentioned above, figure 7 and the surrounding discussion cannot be recreated using the new versions of the parameter functions (primarily, the use of a logit function for the shape parameter), this discussion has been removed.

Page 14 - section 3.5: I am not entirely clear on what is being described here. Why does the assumption that the non-stationary parameters are valid for more than 50 years only hold for 66 stations? Are these stations with more than 50 years or stations for which the $\kappa(t)$ function stays within the required bounds? Do I understand correctly that you are using L=50 and applying the formulae shown in Section 3.3.2.

The original form of $\kappa(t)$ only stayed strictly inside ±1 for 66 stations. The new form stays inside ±0.75 for all stations, so the following text has been deleted: ", assuming that the fitted non-stationary parameters remain valid for the 50 years following the start of the record at the station (66 stations satisfied this)". We obtain non-stationary flood peak estimates by inverting the equation for $T_Q(y)$, as stated on page 9, line 11.

**Anonymous Referee #2**

The paper "Have trends changed over time? A study of UK peak flow data and sensitivity to observation period" by A. Griffin et al. analyses the changes in time of the parameter estimates of the Generalized Logistic distribution and flood quantiles for the flood data of the UK Benchmark Network. The authors use two approaches (i.e. fix-width moving window and fixes-start extending window) to investigate the sensitivity of the parameter estimates to record length and to the presence of most extreme events, under both stationarity and non-stationarity assumptions.

The manuscript is well written, the aim of the paper is clearly stated in the introduction and the analyses and the results are presented in an appropriate way. The methods/approaches are not particularly new, but the results (especially the maps showing the spatial distribution of the trends in the quantiles and parameters) are of clear scientific and technical interest, given that the detection of flood regime changes is a topic of major concern and relevance.

I would nevertheless suggest to the authors some changes concerning mainly the text and the organization of the paper in the result and conclusion section:

The authors thank the referee for their useful comments, and we hope that our responses below reflect that. The conclusion section has been restructured, as have the methods (see below).

Page 1 – Lines 10-11: from this sentence in the abstract it seems that the aim of the paper is to separate the effects of land-use change from climate change. The UK Benchmark Network is used instead to consider near natural catchments only. I would suggest to the authors to rephrase this sentence.

Noted. To reduce ambiguity in the abstract, the sentence has been amended to "… vary through time in the UK. The UK Benchmark Network (UKBN2) is used to allow focus on climate change separate from the effects of land-use change."

Page 1 – Line 29: please define NRFA in the text, I see it is defined at page 3 but you mention it two times before in the text.

The text "National River Flow Archive" has been inserted before the first occurrence of the acronym NRFA (on page 1, line 29).

Page 2 - Lines 8-9: I would clarify in the text that Hall et al. (2014) is a review article. The same at page 4 – lines 8.

Understood. At page 2, line 8, the text has been changed to "Hall et al. (2014) reviewed investigations of flood regime changes from across Europe to identify…." At page 4, line 8, the text is changed to "… the identification of flood-rich or flood-poor periods, as reviewed on a European scale by Hall et al. (2014) may be a strong application for this method."

Page 2 - Line 30: I haven't fully understood the third listed objective. In my opinion it is unnecessary.

The third objective was more heuristic in nature, and was to try and illustrate through example some issues with changing period of record, and more generally in communicating change in flood regime over time. We feel it is worth keeping, but it has been edited to read: "Demonstrate examples of issues regarding the complexities in clearly describing changes in flood frequency estimates".

Page 3 - Line 11: please define AMAX in the text

The words "instantaneous annual maximum" have been inserted before the first occurrence of the abbreviation AMAX on page 3 line 11.

Page 3 - Line 26: please put numbers to the equations

All equations, except those embedded inside paragraphs, are now numbered.

Page 3 - Line 30: I understand the meaning of the sentence but, to be precise, it is not correct to say that T is equivalent to the annual exceedance probability, but rather that they have a one-to-one correspondence according to the given relationship.

This is a good point. The sentence has been changed from "Under stationary conditions $T$ is equivalent to the annual exceedance probability (AEP) where $AEP = 1/T$" to "Under stationary conditions $T$ has a one-to-one correspondence with the annual exceedance probability (AEP), according to $AEP = 1/T$".

Page 4 - Figure 1: It would be helpful to add to this map the locations and the names of the hydrometric stations that are taken as examples later in the manuscript (i.e. the stations of figure 3, 6, 7 and 9). In this way the reader would be able to find also in the maps (e.g. in figure 4) what is discussed at the level of the single station. I would also suggest using different (maybe solid) colors because I find the map not easy to read (the blue and green are very similar).

Labels for the three stations discussed in figures 3, 6 and 9 have been added to figure 1 (note that the new draft of the manuscript removes figure 7). The text "The three stations considered individually in later figures are labelled and outlined in yellow." has been appended to figure 1 caption. In addition, figure 1 has been re-drawn using solid colours in Viridis colour scale D, which is colour-blind and monochrome friendly, and the same colour scale has also been applied to figures 2, 3, and 6.

[Figure]

Available AMAX (years)

- 31 - 39
- 40 - 49
- 50 - 59
- 60 - 77

Page 4 - Line 9: The authors state in section 2.1 that the minimum record length is 21 years therefore isn't this sentence unnecessary?

The qualifier "that has more years of AMAX data than the width of the window" has been moved to page 5, line 2. It is necessary there, as some records have fewer than 40 years of AMAX.

In section 3.1 and (beginning of) section 3.2 of the results (page 4, 5 and 6) the authors mainly describe the moving and extending window approaches, making general considerations and without directly mentioning the results of the study nor the figures. I would suggest revising the organization of these sections (for example by moving the parts that are descriptive of the approaches into the method section) or to refer directly to the figures and results, while describing the analysis. The same applies to section 3.3.1 and 3.3.2 where the authors give definitions of the non-stationary parameters and return periods.

Agreed. Moving all methodological matters to the methods chapter seems logical, and was considered during the initial writing of this manuscript. To this end, sections 3.3.1, 3.3.2, and the start of sections 3.1 and 3.2 have been moved to section 2, and relabelled 2.2.1, 2.2.2, 2.2.3, 2.2.4 respectively.

Page 6 - Figure 3: Why don't the authors plot also the line corresponding to the extending-window in panel e (which is instead mentioned at page 7 – lines 19-20)? I would also mention somewhere in the figure caption that the parameter and Q_MED values are plotted in correspondence of the end year of the moving and extending windows.

The authors thank the referee for drawing this to our attention. We have added an "extending-window QMED" to figures 3 and 6 and the corresponding words into the caption: "Lines on figures (a), (b), (c) and (e) are plotted corresponding to the end of the moving- or extending-windows."

[Figure]

Page 8 - line 16-17: Why do the authors use different methods for parameter estimation in the stationary and non-stationary case? Please provide some explanation for this choice or use the same method for both.

This is a good criticism. See the equivalent comment from referee #1 and our response regarding the change of method to solely using maximum likelihood.

Page 8 - line 19-24: The authors use a linear regression with time for the shape parameter, but a convincing justification for this choice is not given; they highlight instead its negative implications and limitations (also at page 15 – lines 10-12). Please provide some explanation for the choice of this relationship. In agreement with the comment of the Anonymous Referee #1, I believe it's reasonable to try another expression for k(t) that overcomes the current limitations

We have reformulated both κ and α to overcome the limitations associated with linear trends in both parameters. Please see our equivalent response to Anonymous Referee #1 for further details.

Page 9 – line 6: I was not able to find this exact formulation of the return period in Salas and Obeysekera (2014). Is there an assumption about the condition of non-stationarity (increasing, decreasing or shifting extreme events), as done in Salas and Obeysekera (2014)? Can the authors comment a bit more on this definition?

Thank you for pointing out this ambiguity. The expression of $T_Q(y)$ on line 6 has been corrected to be "$T_Q(y) = 1 + \sum \prod (1 - P_Q(s))$", as in the case for increasing extreme events (Equation 8b in the reference). In the present dataset, the probabilities of exceedance were sufficiently consistent, and the return periods sufficiently short to allow the probabilities to converge in such a way that the issues of an event of a given size having an infinite return period due to a decreasing trend (and hence unbounded sum and product) did not arise. To aid clarity, the following sentences on line 7 are added.

"On a technical point, in Salas and Obeysekera (2014), the above definition (defined as an expectation E[X] in the paper) is based on monotonically increasing probabilities of exceedance. However, the same still holds for decreasing probabilities of exceedance as long as they do not decrease or converge to zero too quickly, ensuring that the product term (which equals the probability of at least one exceedance in r years) still produces an appropriate value. These conditions are satisfied in the present dataset due to the relatively short return periods considered and the smaller."

Page 9 – line 9: If P_Q is the annual exceedance probability, as defined at line 7, I believe there is a typing error in the equation. Shouldn't T_Q be equal to 1/P_Q?

Thanks to the referee for spotting this. $T_Q$ is indeed equal to $1/P_Q$ here; we have corrected the equation.

Page 11 – lines 19-20: The authors talk about figure 7 and refer to the stationary estimates that are not shown there. Please add them.

Since, as mentioned above, Figure 7 and the surrounding discussion cannot be recreated using the new versions of the parameter functions (primarily, the use of a logit function for the shape parameter), this figure has been removed.

Page 12 – lines 6-13: I find figure 8 interesting, but I think its description in these lines is bit synthetic and could be improved (only 2 panels out of 12 are actually commented).

We have attempted to extend the discussion around figure 8 and we now refer to trends in each of the GLO parameters at least once. However, it is difficult to extend this discussion further as there are few relationships between trends in GLO parameters and catchment properties. The new discussion (between figure 7 and section 3.4 in the discussion paper) now reads:

"Although this work does not attempt to attribute causes to the trends in UK Peak Flow data, it is of interest to see whether any standard covariates correlate strongly with the trends observed. Figure 8 shows relative change in $\xi$ ($\xi_1/\xi_0$), relative change in $\alpha$ ($\alpha_1$), and $\kappa_1$ against catchment centroid easting, catchment centroid northing, average annual rainfall during 1961-1990 (SAAR) and catchment area. This reveals that the strongest negative trends in $\kappa_1$ are for catchments with SAAR less than about 1000, which are shown to be mostly located in the east of England by co-referencing against the easting and northing subplots. In all cases but one, values of $\kappa_1$ less than -0.03 correspond to positive values of shape parameter initially decaying towards zero, indicating an increasing upper bound on the flood frequency curve as it straightens. In addition, there are no strong trends in $\xi$ for catchments larger than around 500 km$^2$, and $\alpha_1$ is also potentially shown to be closer to zero for larger catchments. In practical terms, this means that the centre of the flood frequency curve is relatively unchanging over time for larger UK catchments. Nothing else conclusive can be discerned from Figure 8, potentially as a result of sampling variability causing some catchments to behave in ways that are not typical, but still plausible given the record lengths involved."

Page 16 - lines 11-12: I believe that this statement about Q_MED is in contrast to what observed in figure 5, panel a, and what stated at page 9 – lines 23-25.

This is well noted, and the authors agree. This statement will be removed in the restructuring of the conclusions section (see below).

Section 4: In my opinion the organization of this section can be improved; it is a bit confused at the moment and, as the Anonymous Referee #1 also comments, there is no clear conclusion or take-home-message emerging. I would be appropriate to refer to the initial objectives, stated at page 2 – lines 27-30, and to re-organize this section accordingly, in order to clearly demonstrate how the analyses in the paper have fulfilled the initial objectives.

Thank you for this comment. To address this comment, and the general comments of Anonymous Referee #1, the conclusions section has been replaced by the following:

[revised manuscript text omitted]

**Final comments**

The authors would like to thank the reviewers for these useful and interesting comments which have led us to rework some aspects of the work, particularly in investigating alternative expressions for the GLO shape parameter, and expanding on some of the less deeply analysed data, such as correlation of trend with covariates and inter-correlation of scale and shape parameters over time.

---

## Author Response (AR2)

[revised manuscript text omitted]

**Changes**

I thank the author for addressing very clearly my comments - the paper has some interesting findings and I believe will be of interest to many.

I have a few additional minor comments:

5 * in Figure 9 the (ratio - 1) is shown for different design events. In the text description of the figure the author refer to the ratio, I believe this should be corrected.

* I take the point the authors make that the logistic link function used is not ideal. I think constraining the shape to be within given boundaries might is useful when one wishes to model it as a function of covariates, but it could surely be done in some other way. I recently saw an alternative link function suggested in

10 https://arxiv.org/abs/1907.04763: this is surely an interesting topic which could be investigated further in the future.

**Responses**

Page 16, Line 4: ratios edited to include ( - 1) to match caption of Figure 9.

Page 4, line 18-19: Comment added - "Other expressions could be used for $\kappa$, see Johannesson et al. (2019) for a more complex

15 example; future work in this direction would be interesting."

Citation added: Johannesson et al. (2019) in references.